# Individual and Geographic Variation in Non-Harmonic Phases of Male Capercaillie (*Tetrao urogallus*) Song

**DOI:** 10.3390/ani13040765

**Published:** 2023-02-20

**Authors:** Richard Policht, Vlastimil Hart

**Affiliations:** Department of Game Management and Wildlife Biology, Faculty of Forestry and Wood Sciences, Czech University of Life Sciences Prague, 165 00 Prague, Czech Republic

**Keywords:** Tetrao, communication, vocal individuality, grouse, vocal signature, geographic variation

## Abstract

**Simple Summary:**

The remaining European populations of the capercaillie are restricted to isolated mountains, and population declines have led to local extinctions across Western and Central Europe. The majority of studies related to individually distinct vocalizations focused on various versions of tonal and harmonic signals, while non-harmonic noisy signals have not been studied in such detail, even though these signals are frequent in some birds including gallinaceous species. We described the structure of capercaillie song and showed how temporal and frequency acoustic variables contribute to individual-specific expression. The combination of temporal and frequency variables showed the best classification result. Capercallie song represents a complex signal of non-harmonic ‘noisy’ sounds formed by different acoustic components organized into four main phases. We tested which song phase makes the largest contribution to coding individual identity. The best contribution to individual variation was found in more complex phases. Recordings from males originating from seven countries also suggest geographical variation underlying capercaillie song. The such geographic variation could reflect the potential genetic differentiation of distant populations. Our results revealed that males from Sweden and Norway (Boreal area) formed a distinct cluster from males in the Czech Republic, Germany, and Poland (Sumava and Carpathian area).

**Abstract:**

Individually distinct acoustic signals, produced mainly as tonal and harmonic sounds, have been recorded in many species; however, non-tonal ‘noisy’ signals have received little attention or have not been studied in detail. The capercaillies (*Tetrao urogallus*) produce complex courtship songs composed of non-tonal noisy signals in four discrete phases. We analyzed recordings from 24 captive male capercaillies in breeding centres in the Czech Republic, Poland, and Germany, and songs from wild males in Sweden, Norway, Finland, and Estonia to test whether a non-harmonic song can encode individual-specific information. We also analyzed the intra-population variation of the male song from three separate areas: Carpathian (Polish and Czech Beskid), Sumava, and Boreal (boreal range of species distribution). Temporal and frequency characteristics can reliably distinguish capercaillies at the individual level (91.7%). DFA model testing geographic variation assigned 91% of songs to the correct area (Carpathian, Sumava, Boreal). The cluster analysis revealed that males from the Boreal area formed a distinct cluster. Our analysis shows clear geographical patterns among our study males and may provide a valuable marker for identifying inter-population dynamics and could help to characterize the evolutionary histories of wood grouse. We discuss the potential use of this marker as a non-invasive monitoring tool for captive and free-roaming capercaillies.

## 1. Introduction

The capercaillie (*Tetrao urogallus*) is both a keystone species inhabiting coniferous boreal forests [1] and an umbrella species indicative of high-biodiversity ecosystems in mountainous regions [2]. Even though this species shows a widespread distribution over an extremely large range from Spain to Eastern Asia and Russia [3,4], relict European populations are restricted to isolated mountains [5], and population declines have led to local extinctions across Western and Central Europe [6]. Twelve subspecies of capercaillie have been described based on morphological traits [4]; however, additional studies analyzing non-morphometric parameters, e.g., song characteristics, could help to confirm or possibly revise these subspecies classifications [7]. European capercaillie populations form two distinct genetic groups, a southern lineage (Pyrenean–Cantabrian and some Balkan populations) and a lineage found in the remaining Eurasian habitats [1]. Genetic variation identified in some populations of capercaillie highlights the need for research on the potential physiological and behavioral differences between these populations, e.g., relating to courtship displays and acoustic communication at lek sites [1]. A detailed characterization of capercaillie songs is lacking, and previous studies have investigated only low-frequency song components [8,9,10] and not considered inter- or intra-population song variation.

Typically, males begin singing 1–2 h before sunrise from tree perches and perform arboreal song displays, then fly to their display territories where they continue song displays from the ground [3]. Males congregate at traditional display grounds (leks) where females mate with one or a few high-ranking males [11]. Capercaillies have a promiscuous mating system where sexes meet for copulation only, after which females nest and provide maternal care independently [3]. Adult males tend to be solitary, whereas young males form small groups. Females are known to flock and female offspring may remain together with female siblings until spring [3]. The intensive effort underlying male song-displays during courtship likely reflects the importance of song for reproductive success, when males strongly compete for good breeding areas and females [12].

Male capercaillies produce songs containing a complex acoustic structure (Appendix A). The song contains four typical phases: clicks, trills, corks and whetting (Figure 1). A series of clicks initiate the song, which then transitions into the trill, followed by the cork, producing a sound similar to that of a cork pulled from a bottle. The final phase, whetting, is formed by syllables of scraping, grinding noises combined with hissing and squealing sound. Songs are produced only by males during courtship. Traditionally, individual-specific distinction based on acoustic characteristics has been achieved in species by analyzing frequency and temporal parameters of predominantly tonal and harmonic sounds, while non-tonal ‘noisy’ features with the un-periodic vibration of vocal folds in the syrinx and non-vocal sounds have not been given adequate attention. Therefore, the role that non-tonal signals may play in individual recognition remains largely unknown [13]. Non-tonal sounds often appear ‘noisy’, and their spectra are not expressed as a fundamental frequency with multiple harmonics. Such sounds can result from a single impulse, e.g., a click, or from inherently stochastic, turbulent airflow [14]. Despite the non-tonal nature of capercaillie courtship songs, we predict that these vocalizations contain unique ‘signatures’ specific to individuals of this polygynous bird species. If so, these acoustic markers could be used for monitoring and management purposes. Indeed, such a non-invasive method would be especially useful for species such as the capercaillie that are extremely sensitive to any form of manipulation, including trapping, handling, and marking [15].

We analyzed recordings of capercaillie songs from males held in breeding centres in the Czech Republic, Poland, and Bavaria, as well as songs from wild males in Sweden, Norway, Estonia and Finland, to test whether atonal, non-harmonic songs with complex acoustic structures can encode individual-specific information. We tested two levels of variation: individual and geographical variation in complex non-learned vocalization produced during mating song display. Genetic factors should be considered as a source of variation in species with non-learned vocalizations [16,17]. Such signals can be additionally designed by ecological and evolutional processes in different ways regarding various acoustic parameters, e.g., the peak frequency and the frequency range of non-learned vocalizations, could be correlated with ecological parameters of the habitat, while the other parameters, e.g., the number of syllables, their structure, and the fundamental frequency, should be rather genetically constrained and thus influenced by syringeal morphology and driven by phylogeny [18]. We can predict that acoustical variables related to morphology and genetic predispositions should play a key role in individually specific songs of this extremely polygynous species, where males intensively compete for access to females, and thus morphological and physiological condition should be expressed in their song display. These parameters should significantly contribute to acoustic variation between different males, which are under both strong selection by females and under high competition among males.

We also expect geographic variation due to the residency and limited dispersion abilities of the capercaillies, especially in actual situations of isolated populations in most areas of central and western Europe. We could therefore expect two main hypotheses regarding the geographic pattern of their song: (1) evolutionary mechanisms following genetic isolation may influence syringeal morphology and thus design mainly acoustic parameters related to syringeal morphology (see above) or (2) alternatively, isolated populations inhabiting different areas should lead to habitat as a key factor in designing acoustic parameters responsible for geographically distinct songs. This acoustic adaptation hypothesis [19] predicts the design of an acoustic signal that is best adapted to local habitat conditions. The question arises whether these hypotheses may also suit non-harmonic broadband acoustic signals with predominant chaos frequency structures, which mostly characterize all phases of the song produced by this wood grouse, where most acoustic parameters quantify the distribution of energy in the frequency domain and where significant energy lies at lower signal frequencies.

We then compared the independent contribution of each song phase to vocal individuality. We can order these phases based on structural complexity, i.e., more complex phases are those containing multi-syllabic components, which are also associated with longer phase durations. For example, the cork represents the shortest mono-syllabic phase; the click is a longer two-syllabic phase, and the trill and whetting are multi-syllabic song phases. The specific temporal and frequency components of each phase can contribute increasing complexity to the song. We predicted that phases with more complex acoustic structures would allow for a higher degree of individual distinctiveness. Therefore, we expected the following phases to contribute to vocal individuality in increasing order: cork, click, trill and whetting, where cork represents the least complex phase, and whetting represents the most complex song phase. However, previous studies have shown that when individual variation within songs approaches or exceeds the variation between individuals, it is not possible to discriminate at the individual level [20].

Furthermore, we evaluated the intra-population variation of the male song within three isolated areas: Carpathian (from Polish and Czech Beskid populations), Sumava (Sumava Mountain population) and Boreal (individuals of several populations located in the boreal area of capercellie distribution), helping to uncouple individual song variation resulting from geographic factors. Lastly, we compared songs from all locations to determine if geographical patterns emerge between song characteristics in males from separate populations.

## 2. Materials and Methods

### 2.1. Recording

All songs were recorded with Olympus Linear PCM LS-5 and ZOOM H5 digital audio recorders in combination with a Sennheiser ME 62 omnidirectional microphone (frequency response 20 Hz–20 kHz ± 2.5 dB) equipped with a K6 powering module and Earthworks QTR (frequency response 3 Hz–50 kHz ± 1.5 dB). Recordings were saved in wav format (48 kHz sampling rate and 16-bit sample size). The distance between focal males and the microphone ranged from 0.5 to 3.0 m. We analyzed 306 songs from 24 adult male capercaillies either from captivity or from the wild. We recorded spontaneous songs from the following males in captivity: nine males from the Capercaillie Breeding Centre in the Wisła Forestry District located in the Beskid Mountains of Poland, one male from the Breeding Centre in Krásná (Krásná, Beskid Mountains, Czech Republic), four males from the Sumava Mountains, Czech Republic, and one male was recorded from Bavaria (Bayerwald-Tierpark Lohberg). Captive males were housed either in pairs or with several females, and no two males were housed together. Songs were recorded during two recording periods between 31 March and 28 May 2016 and 2 and 3 May 2017. These periods coincided with the courting season, and all captive males displayed courtship behaviors throughout the recording sessions; thus, all recorded songs represent courtship songs produced during the courtship season. The remaining nine songs were obtained from recordings available online (https://xeno-canto.org, accessed on 12 May 2018), which included wild males: four males from Sweden, two from Finland, two from Norway, and one male from Estonia. For the analyses, we selected only songs produced by single males that were observed during recordings. These males were recorded between 13 April and 11 May. Males in captivity were distinguished by colored and numbered rings visible from a distance of several meters. In addition, captive males were kept individually, or only with other females, in separated aviaries, enabling us to identify the identity of singing males.

### 2.2. Acoustic Analysis

For acoustic analysis, only clear songs that did not overlap with simultaneous songs from nearby males or any other auditory disturbances were selected. A total of 306 songs (15 songs from one male from Bavaria, 62 songs from four males from the Sumava Mountains, 124 songs from nine males from Poland, 15 songs from one male from the Beskid Mountains, 20 songs from two males from Norway, 48 songs from four males from Sweden,15 recordings from two males from Finland, and 7 songs from one male from Estonia) were selected and analyzed. In order to test geographic variation, we also included recordings of wild birds from Northern Europe. Since wild birds were not individually marked, we could not use a repeated-measurement design. For the analyses, we used songs from different song series. However, although we tried to record captive birds repeatedly, the samples between recording sessions were not balanced as some birds were not always vocal or repeating the recording was not possible. Furthermore, repeated recording is often risky for this extremely shy bird species. The presence of an unfamiliar person may cause an extreme behavior response, which can even lead to the mortality of recorded birds. As the mating season progresses, females also start building nests and incubate clutches, so recording may be risky for successful incubation due to egg damage or nest abandonment. Recording of these birds, which are, in many western European countries, one of the most endangered species, need to fully respect the management of breeding stations; thus, the recording design is not always optimal. The final sample of recordings from captive birds come from two to three recording sessions per individual, with intervals ranging from four days to one year.

After selection, song phases (clicks, trill, cork and whetting) were manually labelled within each recording (Figure 1). Clicks are present in three forms: two-syllabic clicks (here called clicks), mono-syllabic clicks (solo clicks), and the introduction click. An increasing frequency of clicks gives way to the trill, and because clicks can be produced independently and therefore occur outside the full song, an appropriate method had to be established for defining the trill phase. We found that the trill is formed by the quick repetition of mono-syllabic clicks. Therefore, we defined the two-syllabic click located just ahead of the trill (i.e., the last click before the start of the trill) as the ‘introduction click’ and the click preceding the introduction click as the ‘first click’ (Figure 1).

Spectrograms were created with Avisoft SAS Lab Pro (Avisoft Bioacoustics, R. Specht, Berlin, Germany) using the following parameters: FFT length, 1024; frame size, 100%; overlap, 87.5%; Hamming window; frequency resolution, 47 Hz; and time resolution, 2.67 ms. We measured two sets of acoustic parameters quantifying both the (1) temporal (measured in Avisoft) and (2) frequency domains of the songs (measured in Raven Pro 1.5 Sound Analysis Software; Cornell Lab of Ornithology, New York, NY, USA) [21]. Temporal parameters contained song duration, duration of all phases (clicks, trill, cork, whetting), intervals between phases, which included the pre-whetting interval (between the whetting and cork phases), the pre-trill interval (between the trill and introduction phases) and the pre-introduction and click interval (between the introduction click and the first click). In order to capture variations in the trill, we counted the number of intervals between individual clicks that make up the trill and identified three intervals: the first trill interval, the last trill interval and the median trill interval, which covers the largest interval in the middle of the trill (frequently separating several start clicks from clustered clicks). In cases where all clicks of the trill were regularly spaced (i.e., no large or irregular inter-click interval), the center (i.e., median) interval was selected as the median trill interval. We counted the number of click types for each male (i.e., clicks, solo clicks, and introduction clicks). In order to quantify the most complex and longest phase (whetting) formed by a quick series of scraping syllables fluctuating in intensity, making it more difficult to identify and categorize syllables, we selected the longest syllable for measurements (labelled long whetting) and the following shorter syllable (short whetting) and quantified their temporal and frequency parameters as described above. These syllables must not be located at the start or end of the song. We also measured additional temporal parameters in these two whetting syllables: peak time (relative time at which the spectrogram bin with the highest spectrogram level occurs relative to the duration of the signal).

Frequency parameters include the following measurements of waveform: (Q1F) 1st quartile frequency (frequency dividing the signal into two frequency intervals containing 25% and 75% of the energy), (Q3F) 3rd quartile frequency (frequency dividing the signal into two frequency intervals containing 75% and 25% of the energy), (Center F) central frequency (frequency dividing signal selection into two frequency components of equal energy, (BW90) 90% bandwidth (difference between the 5% and 95% frequencies), (F95) frequency dividing the signal into two frequency intervals containing 95% and 5% of the energy, and three quantifications of entropy (maximum, average and aggregate). The entropy quantifies the disorder of the signal by analyzing the distribution of energy. Higher values correspond to greater disorder in the signal, e.g., pure tones with energy in only one frequency bin correspond to zero entropy [21].

### 2.3. Statistical Analysis

We excluded highly inter-correlated parameters with r > 0.80 [22]. The remaining 27 parameters were used in the analyses that combined parameters associated with individual males, resulting in the percentage of correctly classified songs at the individual level [23]. We standardized these variables using Z-score transformations (by subtracting the mean and dividing by the variable’s standard deviation), which avoided the false attribution of weights to acoustic parameters measured in different units (IBM Corp., Armonk, NY, USA). The results of the DFA were validated using a leave-one-out cross-validation procedure (IBM SPSS Statistics 20).

We conducted a series of DFAs to classify songs collected during various subsequent recording sessions (from four days to one year). When we saw that DFA results were stable regardless of which temporal data were used, we pooled recordings from different time periods per individual into the final DFAs.

In addition to individual variation, we also tested geographical variation, but these two factors are not statistically independent. Therefore, we performed permuted DFA (pDFA) for a nested design representing a randomization process used for two-factorial non-independent datasets for cases of one factor nested in another factor by comparing the percent correct revealed in the model to the distribution of percent correct values based on randomly assigning the group identity to each individual. We conducted pDFAs using a script written in R software (provided by Roger Mundry) using 100 random selections and 10,000 permutations. Permutations of DFA enable the calculation of the percentage of correctly classified objects relative to the original (unpermuted) data based on the songs used to derive discriminant functions and the percentage of correctly classified songs for the cross-validated (permuted) data, which were not used to derive discriminant functions [24]. Details of these calculations have been previously described by Mundry and Sommer [24]. The procedure provides a *p*-value in order to determine the significance of the observed correct classification rate of songs to the test factor (e.g., the area in our study) while controlling for a single nested factor (e.g., individual) [25]. Discriminant functions (averaged per individual males) were entered into a hierarchical cluster analysis using PAST (version 3.20) to show how individual males cluster. The clustering method was based on the cophenetic correlation coefficient [26]. We used 1000 bootstrap replicates to show the nodal support of the dendrogram [27].

The original dataset was used for descriptive statistics (306 songs), while datasets used for DFA models did not include songs with any component of the song missing. A random subset of the dataset (102 songs) was independently coded by the second observer to check inter-observer variance, and Spearman correlation ranks showed correlations between observers with r_S_ > 0.9.

Descriptive statistics provide the means ± SE using STATISTICA 13 (Dell Inc., Round Rock, TX, USA, 2016), and discrimination analyses were performed using IBM SPSS 20 (IBM Corp., Armonk, NY, USA). We refer to “N” for the number of males and “*n*” for the number of songs used in each test. In the text, we mention conventional classification results and more robust cross-validated classification results. We also used permuted DFA results (pDFA) when the model contained two factors (individual and area) together (see above). For the review of the literature related to vocal individuality, we used articles (*n* = 149) from a 50-year period (1968–2018), which included any comparison of acoustic signals between individuals (not only DFA but also univariate statistics, etc.). We did not include such articles of playback studies that did not contain acoustic analysis of individually specific signals.

## 3. Results

To demonstrate the degree of individual song variation within males from the same population, we independently analyzed recordings from isolated areas: Carpathian (Czech and Polish Beskid: Wisla, and Krasna), Sumava (Sumava Mountains) and Boreal (Sweden, Norway, Finland, and Estonia). The degree of individual distinctness (I) was tested independently on the area, and we also tested how different types of acoustic parameters (temporal vs. frequency parameters) contributed to potential divergence. Next, we investigated the accuracy with which we can assign each song to the geographic area of sampled populations (geographic variation) and (III) the contribution of each song phase to individual distinctness.

### 3.1. Individual Song Variation

Songs were classified to the correct individual in 93.1–98.2% of cases using models which included both temporal and frequency parameters when geographic areas were analyzed independently (Table 1) (Figure 2, Figure 3 and Figure 4). Using only frequency parameters revealed 79.3–89.9% classification accuracy, and temporal parameters revealed 87.9–92.9% classification accuracy. When we tested (Kruskall–Wallis test: H (2, N = 21) for differences between vocal individuality across separate geographic areas, classification results did not yield any differences for frequency parameters (H = 1.243; *p* = 0.537), temporal parameters (H = 0.059; *p* = 0.971), and frequency and temporal parameters (H = 1.025; *p* = 0.599).

Additionally, we tested classification success in all individuals pooled together from all locations using frequency and temporal parameters. The analysis of frequency parameters revealed the lowest classification accuracy (82.2%) in comparison to temporal parameters (85.9%), and the combination of both frequency and temporal parameters had the best classification success (91.7%). Furthermore, the ability to classify songs to correct individuals remained after controlling for area using permuted DFA (pDFA, *n* = 241, *p* < 0.001).

The lowest classification success was always found when using frequency parameters, while temporal parameters revealed higher success. The combination of frequency and temporal parameters revealed the best classification success. Despite these consistent classification trends, however, a Friedman ANOVA test did not find significant differences between parameters: Sumava: χ2 (N = 5, df = 2) = 1.625, *p* = 0.444; Carpathian: χ2 (N = 9, df = 2) = 5.474, *p* = 0.065 and Boreal χ2 (N = 7, df = 2) = 3.125, *p* = 0.210 in classification success based on parameter-specific model inputs.

### 3.2. Geographic Variation

We used a discriminant function analysis to test for the potential clustering of the songs based on area (i.e., Sumava, Carpathian or Boreal). The resulting model excluded songs with missing measurements, including males with missing cork phases (two males from Finland). A discriminant function analysis assigned 91% of songs to the correct area (N = 21, *n* = 243, Wilks’ lambda = 0.111, *p* ˂ 0.001). The DFA model included eleven significant variables (*p* ˂ 0.001). The first two discriminant functions had eigenvalues > 1, while the first function had eigenvalues > 3 and explained 71.3% of the variation. The first discrimination function strongly correlated with a short whetting duration (r = 0.595) and cork central frequency (r = −0.510), while the second discrimination function correlated with the first trill central frequency (r = 0.505). Songs of males from the Carpathian area reached the best classification accuracy (94.9%). Songs from Sumava were classified with 88.4% accuracy, and the Boreal area with 86.2% accuracy (Figure 5). Randomization procedure confirmed that these results were significant relative to geographic area while controlling for the individual identity (pDFA, *n* = 243, *p* = 0.002).

The cluster analysis (cophenetic correlation coefficient c = 0.80) revealed that males from the Boreal area formed a distinct cluster, while males from geographically closer regions (i.e., Carpathian area and Sumava) partly overlapped. In comparison to all other males from the Boreal area (Norway, Sweden), a male from Estonia was included in the Carpathian cluster (Figure 6).

Additionally, we tested which acoustic parameters in this geographic model significantly differed. Four parameters differed based on Kruskal–Wallis ANOVA (H (2, N = 21): short whetting duration (H = 11.81; *p* = 0.003), first trill center frequency (H = 10.01; *p* = 0.007), first trill agg. entropy (H = 8.501; *p* = 0.014), and cork center frequency (H = 10.10; *p* = 0.006). The following post hoc comparison revealed significant differences between the Boreal and the Carpathian area in three parameters (short whetting duration, first trill agg. entropy, and cork center frequency) and the Boreal and Sumavan areas in two parameters (1^st^ Trill Center Frequency and Cork Center Frequency). Sumava and the Carpathian area did not differ (*p* > 0.05).

### 3.3. Song Phase Analysis

Despite the fact that the composition of capercaillie song is highly conservative and that the order of phases is invariant, there is some space for syntax variation (i.e., the composition of phase variants). Therefore, and unsurprisingly, some components of the song exhibit the capacity for larger variation, while others tend to be more conserved. This section describes the contribution of each independent phase to individual distinctiveness. We ordered these phases based on structural complexity: (1) the cork phase represents the shortest mono-syllabic phase, (2) the click phase demonstrates longer duration forms, predominantly two-syllabic sound, while (3) the trill and (4) whetting phases show multi-syllabic structures. Differences in temporal and frequency parameters, as a function of phase complexity, were identified using Friedman ANOVA in the following acoustic components: duration of phases (χ2 = (*n* = 271, df = 4) = 1069.1; *p* < 0.001), BW90% (χ2 = (*n* = 267, df = 4) = 412.9; *p* < 0.001), and center F (χ2 = (*n* = 267, df = 4) = 548.8; *p* < 0.001).

#### 3.3.1. Clicks

Solo clicks were present in only 4.1% of all clicks and found in only 16.3% of songs analysed. At least one song from 13 males contained solo clicks. Solo clicks were present in 56.5% of males analysed. Because the introduction click and the first click contained only one uncorrelated variable, we could not use DFA; thus, we used univariate statistics. A Kruskal–Wallis ANOVA revealed a significant difference among males in the introduction duration (H (20, *n* = 241) = 170.2; *p* ˂ 0.001) and first click average entropy (H (20, *n* = 241) = 162.9; *p* ˂ 0.001).

#### 3.3.2. Trill

A trill is a more complex phase of the song compared to clicks. The trill phase contained 13 uncorrelated variables (six temporal variables and seven frequency variables). Temporal variables included trill duration, pre-trill, number of clicks in the trill (trill), first trill, last trill and med. trill. Frequency parameters contained first trill BW90, first trill center F, first trill F95, first trill AggEnt, last trill BW90, last trill F95 and med. trill AggEnt. The resulting DFA model (N = 21, *n* = 243, Wilks’ lambda = 0.0001) included nine variables (four temporal and five frequency parameters). This song phase reached an 80.8% (85.4%) classification success. All nine discriminant functions were highly significant (*p* ˂ 0.001). The first two functions had eigenvalues > 2 and described 64% of the variation. The first five functions had eigenvalues > 1 and described 92.5% of the variation. The first function correlated strongly with the last trill interval (r = 0.599), and the second function correlated with first trill F95 (r = −0.281) and last trill F95 (r = −0.267). Individual variation can also be seen on the spectrogram (Figure 7).

#### 3.3.3. Cork

The cork represents the shortest phase of the song. Even though the cork is described using only three parameters (temporal: cork duration, frequency: BW90 and center F), this phase significantly contributes to individual distinctiveness (51% classification accuracy, 55.2%, conventional result). All three variables of the resulting model (N = 21, *n* = 243, Wilks’ lambda = 0.016) were highly significant (*p* ˂ 0.001). The first two variables had eigenvalues > 2 and described 86% of the variation. The first function correlated with center F (r = −0.734), and the second function correlated with cork duration (r = 0.620).

#### 3.3.4. Whetting

Whetting represents the longest and most complex phase of the male capercallie song. The following eight parameters were used for the analysis: five temporal (long whetting duration, short whetting duration, whetting duration, pre-whetting and long whetting peak) and three frequency parameters (short whetting, q3f and long whetting max ent). This discrimination model (N = 21, *n* = 241, Wilks’ lambda = 0.0001) showed the second-best classification success with 79.3% (82.6%) classification accuracy. All five discriminant functions were highly significant (*p* < 0.001). The first two functions had eigenvalues > 6 and described 83.7% of the variation. All five functions had eigenvalues > 1. The first function correlated with long whetting whetting duration (r = −0.836), and the second function correlated with short whetting duration (*r* = 0.758)). Individually specific pattern can be assessed on the spectrograms of the whetting (Figure 8).

## 4. Discussion

Individually distinct vocalization has been documented in 22 bird orders based on articles focused on acoustic analysis (playback studies without acoustic analysis were not included). Most research has been related to passerines (42%), and the majority of orders (*n* = 12) comprised less than 2% of these studies. Three orders represented less than 3%, and another three orders represented 3–7% of published papers related to vocal individuality. Strigiformes represent the most frequently studied non-passerine order (14%), and Galliformes were studied in 8% of papers. The majority of studies focus on various versions of tonal and harmonic signals. On the other hand, non-harmonic and noisy signals, including non-vocal sounds, have not received much attention or have not been studied in such detail (less than 4% of all articles), even though such signals are frequent in some bird taxa, e.g., Tetraonidae, Otididae or Anserinae.

Here we describe the structure of capercaillie song and show how temporal and frequency parameters contribute to individual-specific expression. We have found that using only temporal or frequency parameters is sufficient for individual recognition: temporal parameters (85.9% cross-validated classification result) and frequency parameters (82.2%) for males from all areas. The combination of both temporal and frequency parameters revealed the best classification result (91.7%). These results are consistent across all three areas (Sumava, Carpathian, and Boreal areas), and the ratio of correctly categorized songs did not differ between these areas. Using only one type of acoustic parameter (either frequency or temporal) provides comparable classification success with other members of the order Galliformes: 83.3% classification success in great curassow [28], 89.3% classification success in horned guans [29], and 70% and 90% success in female and male peafowls, respectively [30]. Individual-specific differences have also been shown in the non-vocal wing-beating drumming of ruffed grouses (74.6% classification success) [13]. However, the use of both temporal and frequency parameters together results in a higher classification result in our study. It is noteworthy that all studies mentioned above used combinations of temporal and frequency parameters. Only one study presented results for frequency and temporal parameters independently. These results come from the non-vocal wing-beating drumming of the Ruffed grouses; when frequency parameters revealed a 45.5% cross-validated result, temporal parameters had 64.1% success, and the combination of both yielded the best result (74.6%) [13]. Such results from non-vocal wing-beating of ruffed grouses show a similar pattern as those ‘noisy’ non-harmonic songs of capercaillies. Results revealing ~92% classification success (using frequency and temporal parameters) in non-harmonic signals is surprising given that more common parameters, such as fundamental frequency, which are frequently related to information coding, including individual identity, body size, sex, etc., were not considered in the current study. Capercallie song exhibits complex signals of non-harmonic ‘noisy’ sounds formed by different acoustic components organized into four main phases. We tested which song phase makes the largest contribution to coding individual identity. We expected greater benefit from more complex phases, which provide larger acoustic space for encoding information. Less complex initial phases, i.e., clicks, had only one uncorrelated variable, so we could not use DFA to test classification performance. Although the cork represents the shortest and the least complex phase of the song, it contributed significantly to individual distinctiveness and had a classification accuracy of 51% (55.2% conventional DFA). Trill phases, which are more complex than corks, revealed the best classification success at 80.8% (85.4%) and had higher classification accuracy than the most complex phase, whetting, at 79.3% (82.6%). Generally, the best classification success was found in more complex phases (i.e., trills and whetting) and is consistent with our predictions. The best classification result of trills is surprising in comparison to the more complex whetting, but the difference is relatively small (1.5% of cross-validated results). The lower number of uncorrelated parameters in whetting entered into discrimination models could help to explain the difference in classification performance. Nonetheless, clearly, this is a positive correlation between song phase complexity and the ability to identify individual capercaillie from songs.

Recordings from males located in seven different countries suggest geographical variation underlying capercaillie song. Surprisingly, males from Finland failed to produce the cork phase in their songs, which may be a unique characteristic of male songs from this population or may be a result of sampling bias drawn from a small sample size (i.e., only two individuals analyzed from Finland). The absence of the cork phase has also been described in males from the eastern end of their natural ranges in Russia [3]. Songs could be assigned to the correct area in 91% of cases (94.9%—Carpathian area, 88.4%—Sumava, and 86.2%—Boreal area). Thus, geographic variation could reflect the potential genetic differentiation of distant populations, resulting in differences in male songs. Genetic factors likely influence the songs of capercaillie more than learning, which has been shown to influence the acoustics of other non-passerine birds [31,32]. However, it is possible that captive birds, such as the subset of birds used in this study, may indeed exhibit some degree of song sharing, as these birds were held in small and stable groups, providing ample opportunities to encode songs produced by neighboring males. Changes in song quality to match songs of neighboring males would indeed suggest that modification of song according to social surroundings plays a role in capercaillie acoustic behavior and cognition. If we assume that singing has a functional role in both male competition and/or female choice, matched songs would be more easily detected by listeners, thus providing an individual benefit during such interactions. Conditions in captivity where several males sing in proximity might be analogous to situations on the lek. Potential song matching could lead to the higher similarity of neighboring males in comparison to males from other areas. Thus, the divergence of wild populations could be shaped by such a process and parallels findings from captive males in close proximity. Besides known examples of learned vocalization in birds that have been documented in three clades (songbirds, parrots and hummingbirds [33], some ability of vocal flexibility has been shown in several non-passerines. For example, loons change their territorial calls (yodel) when they change territory [34], and mallard ducklings and adult crested tinamous increase peak frequency and call amplitude in situations of increased background noise [35,36]. We cannot exclude a potential modification of the capercaillie songs as an effect of their social and captive environment. Future work is needed to support or refute this idea.

There is also the potential possibility to compare the song of grouse living in captivity and in the wild. Although such a comparison shows a potential difference in half of the measured acoustic parameters, the problem is that the influence of geographical variability cannot be filtered out since all recordings of wild birds come from the boreal region of Northern Europe (Finland, Sweden, Norway and Estonia), and all other individuals were bred in captivity. For this comparison, it would be necessary to record both categories of birds belonging to the same area. Northern Europe offers a good potential to obtain recordings of grouse living in the wild, but there is practically no possibility of recording birds in captivity. For this comparison, the Polish population with a good breeding tradition in captivity could be used for such purposes.

Potential vocal divergence across distant populations could follow genetic differentiation caused by geographic isolation and inter-population distances. We have found a clear vocal distinctness based on geography. A clustering method revealed that males from Sweden and Norway (Boreal area) formed a distinct cluster from males in the Czech Republic, Germany and Poland (Sumava and Carpathian area). Males from Bavaria clustered with Sumava, and males bred in Krasna (Czech Beskid) clustered with other Carpathian males from Polish Beskid. Surprisingly, a male from a continuous boreal distribution in Estonia revealed similarity with Carpathian males. This could reflect the higher similarity of “continental” populations (Baltic countries) to central European countries despite the fact that these populations inhabit continuous boreal taiga as well as populations from Fennoscandia; however, they are geographically separated by the Baltic Sea. Such speculation needs to be verified by more robust sampling.

Previous molecular studies have shown that European populations form two genetic groups, a southern lineage (Pyrenean–Cantabrian and several Balkan populations) and a boreal lineage (remaining Eurasian areas) [7]. The populations which we studied belong to the boreal genetic lineage. Including males from clearly genetically distinct populations, especially from the southern lineage, would show how independent evolutionary divergence was followed by the divergence of capercaillie song. We suppose that the expansion of this study with a more detailed understanding of vocal differentiation across the capercaillie range could help in a much-needed revision of subspecies taxonomy, which has been formed primarily on morphological characteristics and shows obvious polyphyly [1].

Comparison of the level of individual versus geographic variations yielded similar results. The best classification results were achieved using the combination of temporal and frequency parameters. Exclusivity of one kind of acoustic parameter (designed by ecology or morphology/genetics) for both variations (individual vs. geographic) was not confirmed according to our predictions. We suppose that the complex structure of this song in a highly modulated broadband frequency pattern could be designed for both individual morphology (including genetics and ontogeny) and habitat conditions. Individual variation was significantly influenced by temporal parameters (e.g., temporal pattern of trill elements, duration of whetting elements, and cork duration) and frequency parameters (frequency 95 in trills, cork center frequency). Similarly, geographic variation was also loaded by both types of parameters, mainly the duration of some whetting elements and central frequency for the trill and cork elements. This highly complex broadband song is probably designed by strong sexual selection. The conspicuous acoustic display of capercaillie males probably reflects a significant adaptive function during courtship behavior. The most intuitive purpose is to attract females for mating, which likely offsets the costs of energy expenditure associated with the elaborate display as well as increased predation risks [4,12].

The most accurate estimation of the population depends on the monitoring method used. For Capercaillie monitoring, a variety of methods have been used, e.g., transect surveys [37,38], genetic methods [39,40], and the monitoring of displaying males at leks [41,42,43], including the automated acoustic recording of males on leks [44,45]. This bioacoustical method offers the long-term monitoring of male activity on leks. The method of non-invasive acoustic monitoring of grouse birds based on individual-specific voices has already begun to be considered an advantageous monitoring method [46]. A semi-automated call analysis enables species recognition among all other sounds. Such monitoring techniques can reduce disturbance and observer biases [44]. Our results would enable expanding this method from species recognition to the individual level and thus allow the long-term monitoring of specific individuals.

## 5. Conclusions

The capercaillies have a lek mating system, in which males display to indicate their breeding condition. During their song display, males produce complex courtship songs composed of non-tonal noisy signals in four discrete phases. We showed the level of individual variation using (1) only temporal parameters, (2) frequency parameters, and (3) a combination of temporal and frequency parameters. We tested it independently in the whole song and each song phase. These songs also vary geographically. Songs produced by males from a Boreal area were more distinct than songs of males from geographically closer regions (Carpathian area and Sumava mountain). The songs of the capercaillies thus may provide a valuable marker for identifying inter-population dynamics. Such variation could be helpful as a marker for monitoring wild and reintroduced males originating from captivity.

## Figures and Tables

**Figure 1 animals-13-00765-f001:**
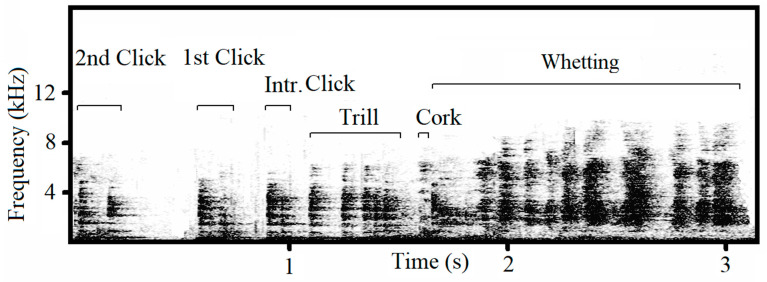
Spectrogram presenting the structure of capercellie song. The duration of the whetting phase shown is truncated. The trill is formed by rapid repetition of mono-syllabic clicks and preceded by the introduction click (two-syllabic click). The clicks are numbered in order from their position relative to the trill (i.e., reverse order from temporal sequence). Whetting is the longest phase containing a rapid series of scraping syllables with irregular noisy acoustic structure (Appendix A).

**Figure 2 animals-13-00765-f002:**
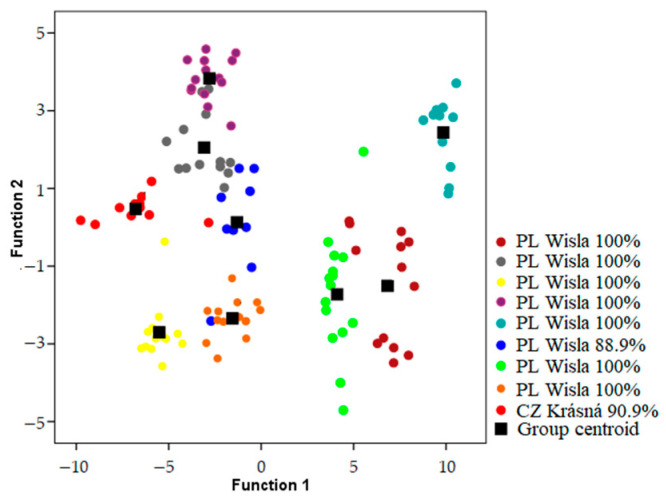
Locations of songs in the plot of the first two discriminant functions recorded from males in Carpathian area and percentage of correct classification of each male. Recordings include one male held in the Krasna Breeding Centre (Beskid Mountains), originally from Poland (Breeding Centre in Wisła Forestry, Beskid Mountains).

**Figure 3 animals-13-00765-f003:**
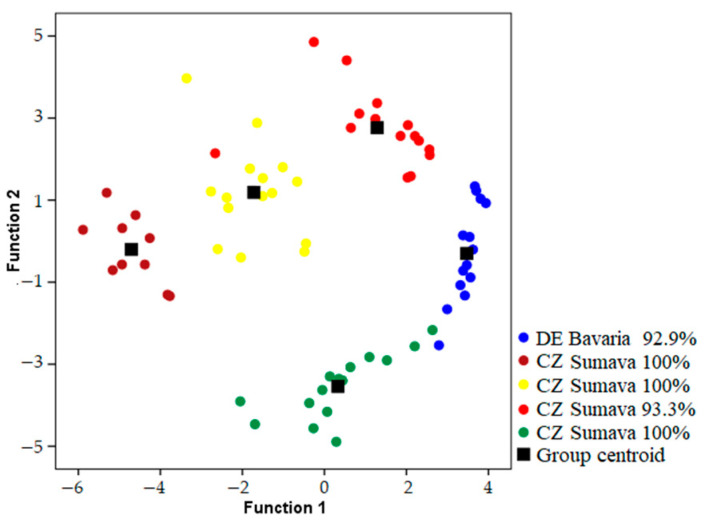
Locations of songs in the plot of the first two discriminant functions recorded from males in Sumava area and percentage of correct classification of each male. Recordings also include one male from Bavaria.

**Figure 4 animals-13-00765-f004:**
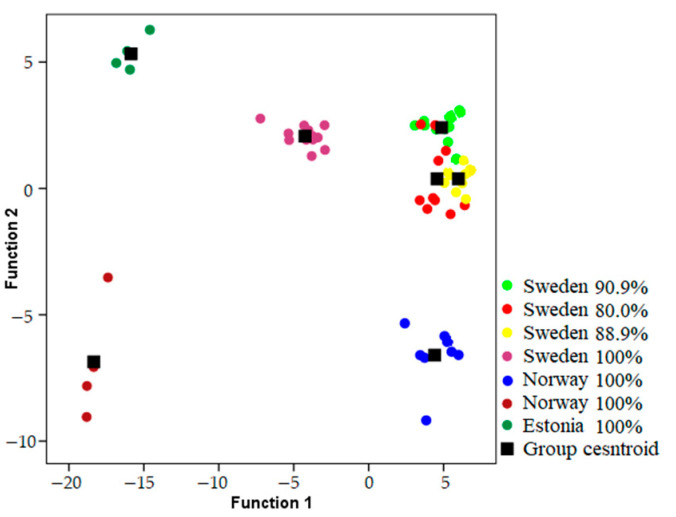
Locations of songs in the plot of the first two discriminant functions recorded from males in Boreal area and percentage of correct classification of each male.

**Figure 5 animals-13-00765-f005:**
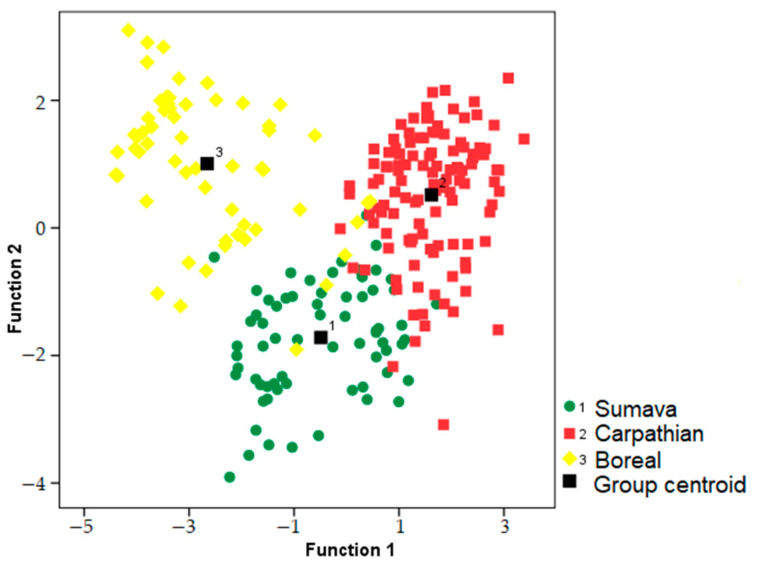
Locations of the capercaillie songs based on DFA to test differences among populations. These two first discriminant functions described 87.4% of variation, and the model classified 91% of songs to correct area.

**Figure 6 animals-13-00765-f006:**
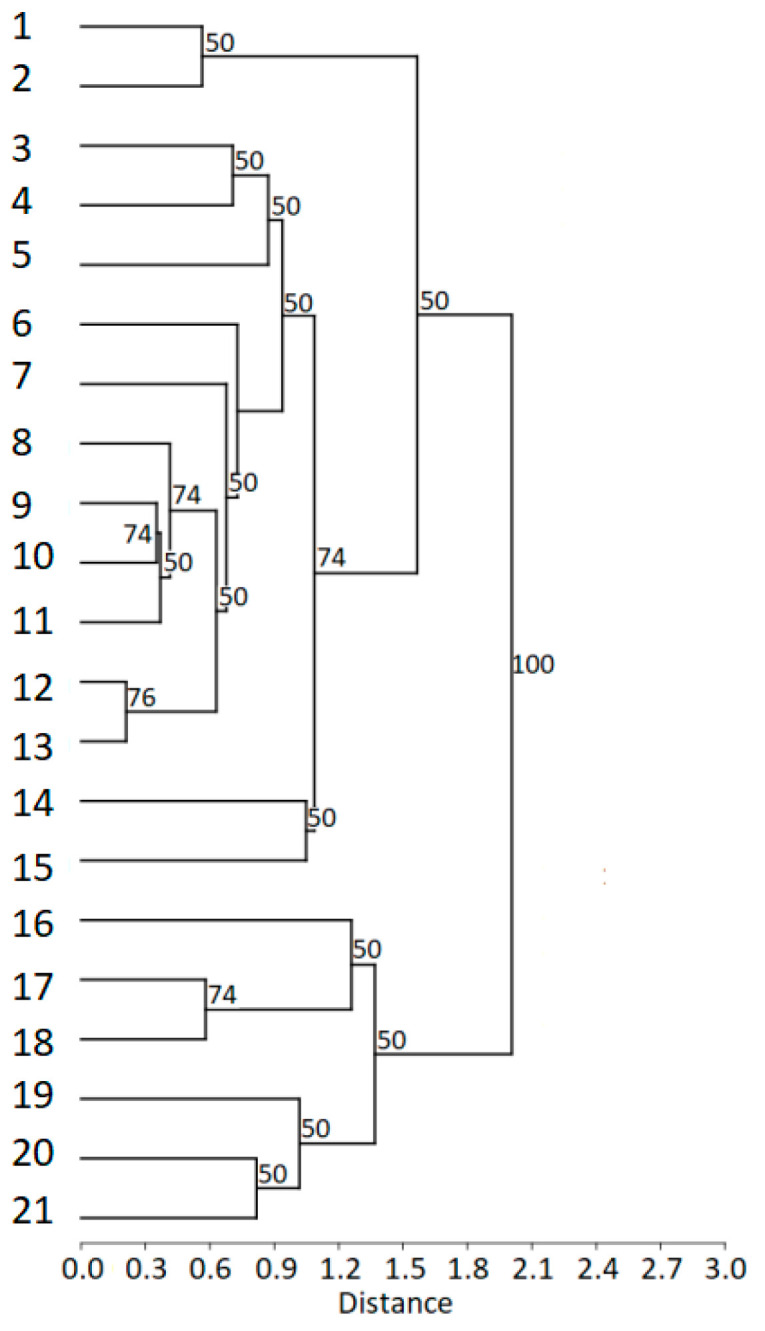
Hierarchical cluster analysis. Dendrogram of individual males created by the single linkage method, based on Euclidian distances (c = 0.80). Bootstrap values show the percentage of replicates (*n* = 1000) where each node is supported. Origin of males: (1) Bavaria, (2–5) Sumava, (6) Estonia, (7–8) Wisla, (9) Krasna, (10–15) Wisla, (16) Norway, (17–18) Sweden, (19) Norway, (20–21) Sweden. Males of Boreal area formed a distinct cluster, while geographically closer males from Carpathian area and Sumava confirmed closer similarity. A male from Estonia showed similarity with Carpathian rather than Boreal males.

**Figure 7 animals-13-00765-f007:**
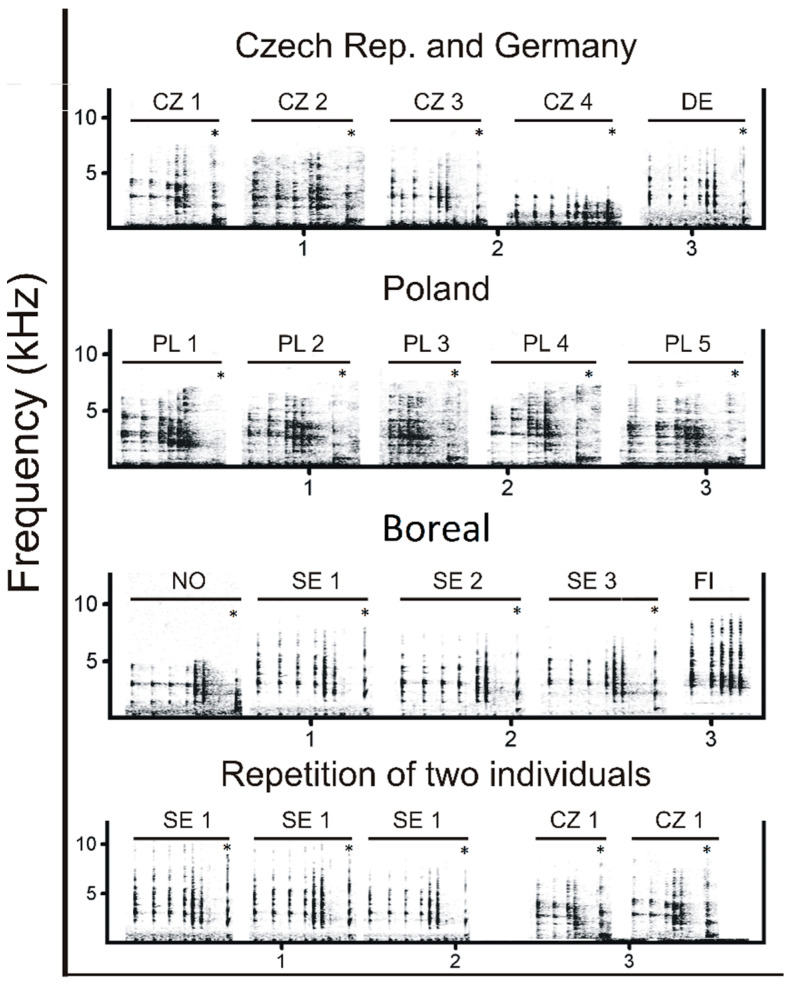
Trill phase variation. A trill is formed by a quick repetition of mono-syllabic clicks followed by a cork (marked by *). Time pattern of these signals significantly contributes to individual and geographic variation. The first row shows trills produced by five different males, four from Czech Republic and one from Germany (all from Sumava area). The second row represents trills from five different males originating from Poland (Carpathian area). The third row shows trills produced by five different males from Boreal area: NO (Norway), SE (Sweden), and (FI) Finland. Males from Finland do not produce corks. The fourth row depicts three repetitions of the same male from Sweden and two repetitions of another male from Sumava.

**Figure 8 animals-13-00765-f008:**
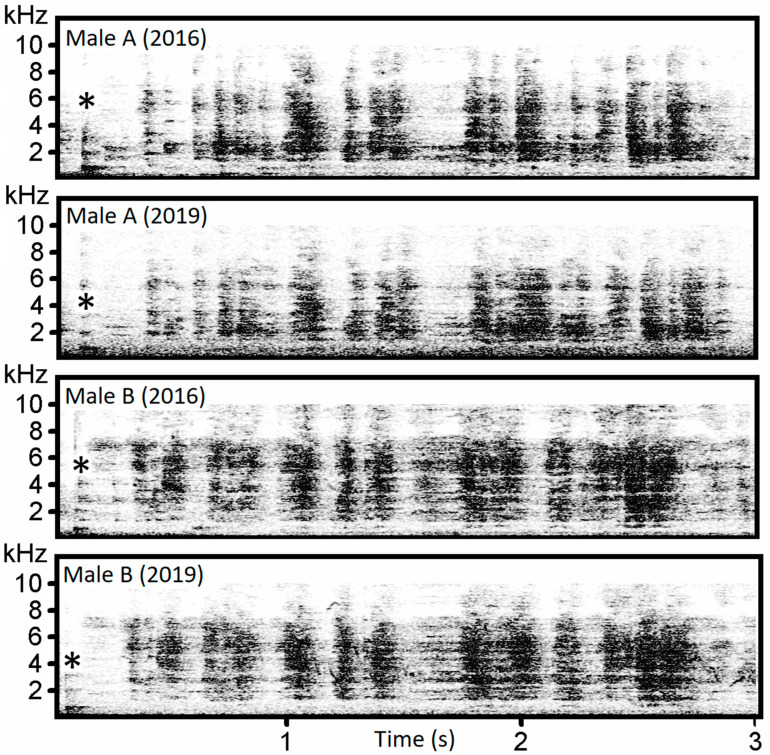
Whetting phase variation. Comparison of whetting produced by two males (A and B) after three years (recorded in April 2016 and 2019). Star shows location of cork, which precedes whetting.

**Table 1 animals-13-00765-t001:** DFA models based on combinations of temporal and/or frequency parameters testing for individual variation.

	Type of Acoustic Parameters Entered into DFA		
	Temporal	Frequency	Temporal + Frequency
**Sumava**	**92.8/92.8%**	**89.9/95.7%**	**97.1/97.1%**
	*n* = 69, N = 5	*n* = 69, N = 5	*n* = 69, N = 5
	var entered = 14	var entered = 13	var entered = 27
	var res = 2, *p* < 0.001	var res = 8, *p* < 0.001	var res = 3, *p* < 0.001
	Wilk = 0.027	Wilk = 0.007	Wilk = 0.001
**Carpathian**	**92.9/94.7%**	**80.7/83.3%**	**98.2/98.2%**
	*n* = 113, N = 9	*n* = 114, N = 9	*n* = 113, N = 9
	var entered = 14	var entered = 13	var entered = 27
	var res = 4, *p* < 0.001	var res = 11, *p* < 0.001	var res = 7, *p* < 0.001
	Wilk = 0.001	Wilk = 0.001	Wilk = 0.001
**Boreal**	**87.9/91.4%**	**79.3/82.8%**	**93.1/94.8%**
	*n* = 58, N = 7	*n* = 58, N = 7	*n* = 58, N = 7
	var entered = 14	var entered = 13	var entered = 27
	var res = 3, *p* < 0.001	var res = 5, *p* < 0.001	var res = 6, *p* < 0.001
	Wilk = 0.0001	Wilk = 0.005	Wilk < 0.001
**All areas**	**85.9/89.6%**	**82.2/86.7%**	**91.7/95.4%**
	*n* = 241, N = 21	*n* = 241, N = 21	*n* = 241, N = 21
	var entered = 14	var entered = 13	var entered = 27
	var res = 8, *p* < 0.001	var res = 9, *p* < 0.001	var res = 11, *p* < 0.001
	Wilk < 0.001	Wilk < 0.001	Wilk < 0.001
	*p*(pDFA) < 0.001	*p*(pDFA) < 0.001	*p*(pDFA) < 0.001

Percentage of correctly classified songs to correct individual is bold (cross-validated/conventional DFA), *n* = songs, N = individuals, (var entered) number of variables entered into the DFA as initial predictors, (var res) number of variables and their significance included in resulting DFA model, Wilk = Wilks’ lambda. Analysis including all areas forms two factors, “individual” and “area”. These factors are not independent when individual is nested in area, therefore results of these DFAs were verified using permuted DFA for nested design; see [24]. Factor “individual” was tested when permutations were restricted to happen within “area” as restricted factor (10,000 permutations).

## Data Availability

The data that support the findings of this study will be openly available in any publicly accessible repository, such as Dryad, as soon as this manuscript is accepted.

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
