# Peer review of "Individual and Geographic Variation in Non-Harmonic Phases of Male Capercaillie (Tetrao urogallus) Song"

_animals, 2023, doi:10.3390/ani13040765_

Round 1

Reviewer 1 Report

Main comments:

This is an interesting study about variation in a complex vocal signal of an understudied species. As such, I think it could make an important comparative contribution to the literature on geographic variation in signaling systems. I have one major concern that can potentially be addressed if the authors have sufficient sampling. If not, I think the 'individual distinctiveness' question cannot be answered by these data.

The authors must provide more information on the range of sampling dates for each individual recorded. For the question of potential individual distinctiveness in signals, it is essential to record signal samples multiple times for each known individual, and over wide ranges of time. As one example, Ellis (2008, Condor 110:648-657) found a strong negative relationship between DFA accuracy of acoustic elements in calls of magpie jays (Calocitta formosa) and the number of days recorded for each individual. Ellis suggested that single sampling days of recordings for each individual can confound motivational or other idiosyncratic influences on signaling with individual distinctiveness. Only multiple sampling periods per individual over wide time ranges can help reduce the likelihood that such motivational influences might account for any differences seen, assuming that those motivational influences likely change over days/weeks/months, revealing only individual distinctiveness given sufficient sampling.

Minor comments by line number:

27. Perhaps include some of the other common names for this species – e.g., wood grouse, etc.

53. I suggest changing “Male capercaillies provide a bird model with songs containing complex . . .” with “Male capercaillies produce songs containing a complex . . .”

54. I suggest dropping one of the ‘typical’ uses here.

57. I suggest the authors describe the mechanism of the whetting component – is this related to feather or other body part scraping?

72-74. An obvious comparison raised here is whether captive songs differ from wild songs, regardless of population of origin. Did the authors look at this?

81-88. This section is the closest the authors seem to get to explicit hypothesis testing. I wonder if it is possible to do this in a more formal way. For example, there are hypotheses about the loss of species-typical behavior due to captivity that go far back in the history of ethology. There is the obvious population-level variation hypothesis (due to phylogeny? adaptation to local environments?). Given the individual- vs group-distinctiveness trade-offs for signal variation, another hypothesis might be that greater individual (or population distinctiveness) should result in lower levels of population (or individual) distinctiveness. There could be other potential hypotheses.

111-114 and 117-120. I really worry the authors cannot differentiate between individual distinctiveness on the one hand and possible motivational influences on signal variation (due to lack of recording over long time periods) on the other hand. See the above-mentioned Ellis (2008) article.

132-134. From the authors’ descriptions, it sounds like these song components are really quite acoustically distinct from one another. Still, it would be good to have another individual independently code a subset of the songs, to obtain (and report!) inter-observer reliability statistics for data coding.

185-223. What happens if the authors use an approach that does not aim to ‘force’ the algorithms to find individual distinctiveness, like DFA does here? If the authors use factor analysis or PCA and then assess whether individuals – or populations – differ from one another based on those scores, do they get similar results?

Results section – without information on the extent to which individuals were recorded over long periods of time, all of the interpretations about ‘individual distinctiveness’ are called into question.

Figures 2, 3, and 4. Some of these ‘population’ graphics are based on recordings of one individual? If so, the ‘population’ distinctiveness question suffers from the same confound as the 'individual' distinctiveness question raised above, at least for those populations with N=1 or 2 individuals recorded.

310-311. Reference to ‘one male from Bavaria’ – is that a mistake?

Figure 5 and associated statistical analyses. I realize it would greatly reduce total information, but if only the mean function 1 and function 2 scores for each male are included in this graphic, what do the results look like?

Figure 6. If some of these ‘locations’ are due to a single male recorded on a single day or two, the same confound as mentioned above enters into this analysis and visual presentation.

Discussion. Everything related to ‘individual distinctiveness’ is called into question for reasons above. The results for population-level differences may still be solid (lines 473-523), if not strong, if the number of individuals recorded for each population is high.

Author Response

We thank the opponent very much for the comments that we tried to incorporate and we hope to improve the value of our article.

Point by point answers can be found in the attached document.

Reviewer 2 Report

The scientific research and the prepared publication are very interesting. They concern a species of bird which is threatened with extinction in many countries. This further improves the scientific level of this manuscript.

The Introduction chapter adequately introduces the reader to the main topic and ends with the properly set goal of scientific research. I wonder if this chapter could be a bit shorter though.

The material and methods used in the experiment were described and explained in great detail. In my opinion, this is a very good point.

Both the Results chapter and the Discussion chapter are prepared and written accordingly.

In conclusion, in my opinion, the manuscript is of high scientific quality, is well prepared and should be published in Animals journal.

Author Response

We thank the opponent very much for the comments that we tried to incorporate and we hope to improve the value of our article.

Round 2

Reviewer 1 Report

I think the authors have done a great job addressing my concerns on the original manuscript. I am still hoping to learn more about potential individual variation over short vs long inter-sampling intervals. The authors write that:

"The final sample of recordings from captive birds come from two to three recording sessions per individual with interval ranging from four days to one year." (lines 171-172)

Can the authors speak to the extent of a possible 'motivational confound' (going back to the Ellis study) by, say, correlating coefficients of variation in acoustic parameters and extent of time between subsequent recording sessions for individuals? This could be really valuable information not just for this system, but for others interested in questions about individual distinctiveness in vocal signals. If they find a strong correlation between those two measures, it seems to me they need to dampen down the 'individual variation' question here a bit. But, if they find ~ no correlation, that would certainly help back up their arguments.

All in all, a really nice paper on an under-studied system!

Author Response

We really thank the reviewer for the recommendations that significantly contributed to the manuscript's quality.

We wanted to record birds repeatedly but that was very problematic due to the responses of birds to the presence of an unfamiliar person during repeating recordings. Our final dataset, therefore, contained repeated recordings in a broad range of time intervals (four days to one year) but these intervals are very different and unbalanced (e.g. three males -1 year interval, 5 males - 1-month interval) that was not possible to use repeated testing.

We added the following part under Methods:

 “To see the stability of DFA results based on recordings from different time sessions, we used a series of DFAs loaded with data from successive recordings (each DFA included data from another different recording sessions). When we saw that DFA results were stable regardless of which temporal data were used, we pooled recordings from different time periods per individual into final DFAs.”

We consider the stability of mentioned DFA classifications as an indicator of individually specific patterns across different time intervals, although we did not succeed to obtain enough data for repeated measurement tests.

Such a pattern indicates a comparison after three years intervals but we have two males only. Spectrograms of these males were included.
